# Bacterial Spore-Based Delivery System: 20 Years of a Versatile Approach for Innovative Vaccines

**DOI:** 10.3390/biom13060947

**Published:** 2023-06-06

**Authors:** Rachele Isticato

**Affiliations:** 1Department of Biology, University of Naples Federico II, Complesso Universitario Monte S. Angelo, Via Cinthia 4, 80126 Naples, Italy; isticato@unina.it; 2Interuniversity Center for Studies on Bioinspired Agro-Environmental Technology (BAT Center), 80055 Naples, Italy

**Keywords:** bacterial spores, spore display system technology, mucosal vaccine, spore engineering, *Bacillus subtilis*

## Abstract

Mucosal vaccines offer several advantages over injectable conventional vaccines, such as the induction of adaptive immunity, with secretory IgA production at the entry site of most pathogens, and needle-less vaccinations. Despite their potential, only a few mucosal vaccines are currently used. Developing new effective mucosal vaccines strongly relies on identifying innovative antigens, efficient adjuvants, and delivery systems. Several approaches based on phages, bacteria, or nanoparticles have been proposed to deliver antigens to mucosal surfaces. Bacterial spores have also been considered antigen vehicles, and various antigens have been successfully exposed on their surface. Due to their peculiar structure, spores conjugate the advantages of live microorganisms with synthetic nanoparticles. When mucosally administered, spores expressing antigens have been shown to induce antigen-specific, protective immune responses. This review accounts for recent progress in the formulation of spore-based mucosal vaccines, describing a spore’s structure, specifically the spore surface, and the diverse approaches developed to improve its efficiency as a vehicle for heterologous antigen presentation.

## 1. Introduction

The COVID-19 pandemic, with 359 M infected people and claiming 5.6 M lives, has firmly confirmed the crucial role of vaccines for human health. 

Since their discovery at the end of the 18th century with Jenner’s and then Pasteur’s studies, vaccinations represent the only way to prevent infectious diseases and induce herd immunity. A long list of prophylactic strategies containing live, attenuated, or killed viruses or bacteria and has led to the suppression or decrease in the incidence of several fatal human diseases. Subunit vaccines, including only a pathogen’s antigenic components able to elicit effective immune responses, such as a polysaccharide [1], a nucleic acid, or a protein, have been proposed as vaccines [2]. So, each year, more than 5 million lives are saved by implementing yellow fever [3], hepatitis B, *Haemophilus* influenza type b, diphtheria, polio, and tetanus vaccines [4].

The current pandemic has highlighted that the technologies available today enable new vaccines to be rationally designed, considering the specific disease. Still, at the same time, it has pointed out some difficulties, such as supply chain issues, storage requirements, and the high cost. Therefore, public health organizations urge the scientific community to overcome these obstacles and develop a new generation of safe, efficient, and cost-effective vaccines. For instance, the World Health Organization has attempted to end cholera by 2030 by developing lower-cost alternative oral cholera vaccines [5].

One of the biggest challenges in vaccine design is developing an efficient product that provides systemic immunity and induces protective mucosal immune responses to block pathogens at their entry site. Inducing the synthesis of secretory IgA (sIgA) at the mucosal surface, an ideal vaccine should elicit immune protection at mucosal surfaces and provide systemic immunity. 

Unfortunately, low levels of mucosal immunity are generated when an antigen is delivered systemically, as in traditional vaccines, but this is not the case when an antigen is delivered orally or nasally [6]. Besides enhancing local immunity, vaccine administration by the oral or nasal route is a non-invasive approach, with improved patient compliance, elimination of the needle-associated risk of transmission of blood-borne diseases, and bypassing of the hepatic first-pass drug metabolism [6]. However, despite these potential advantages, the mucosal route for vaccine administration is still limited. Currently, only nine mucosal vaccines have been approved for use in humans, and they all are live attenuated or whole-cell inactivated vaccine formulations (Figure 1). 

This low number is mainly because of the greater tolerability of orally administered whole-cell killed antigens, antigens’ instability and susceptibility to proteases or endonucleases, and, crucially, the lack of an efficient mucosal delivery system. Therefore, identifying safe and effective mucosal adjuvants conjugated to innovative antigen delivery strategies is critical to advancing mucosal vaccines [13]. The rationale and challenges for developing mucosal vaccines are presented in Figure 2.

So far, several live microorganism display systems, including activated or live attenuated viruses [14], and bacteria [15,16,17], have been considered for developing mucosal vaccine delivery systems. However, each of these platforms has its advantages and disadvantages.

Historically, the first viral vector vaccine was developed almost forty years ago, using as a platform the vaccinia virus to express the hepatitis B surface antigen and provide protective immunity to chimpanzees against the viral disease [18,19,20].

Since then, several viruses, including poxviruses, adenoviruses, and herpesviruses, have been proposed as antigen platforms for vaccine design [18,21,22], and some of them have successfully been used in many veterinary species to protect, for example, against wildlife rabies [23], and new Castel disease [24]. However, only a single viral carrier has been authorized for human vaccination for the Ebola virus [25].

Recently, an innovative vaccine was proposed, based on adenoviral vectors expressing the COVID-19 spike-1, nucleocapsid, and RNA-dependent RNA polymerase antigens. The nasal administration of the new multivalent vaccine showed the ability to provide protective local and mucosal immunity against both the ancestral SARS-CoV-2 and two variants in a murine model [26].

The main advantages of the viral delivery system over traditional vaccines are the high stability and their ability to generate a solid adaptive immune response without an adjuvant [27]. Moreover, viral vectors ensure protection from gastrointestinal tract conditions when orally administrated, and effective release is addressed. Nevertheless, the main drawback remains their safety: inactivated viruses are considered safe but have weak immunogenicity. At the same time, a live attenuated viral carrier shows a greater efficacy but low safety since they could revert toward virulence [28]. In addition, oral viral vaccines have led to post-vaccination complications in some cases. For example, a rotavirus vaccine has increased the risk of intussusception in children, limiting its use [11,29,30].

As viruses, live recombinant bacteria are applicable for oral vaccination since they can induce both systemic and mucosal immune responses [31,32], have intrinsic adjuvant properties [33], protect antigens during the gastrointestinal passage [34], and finally, are easy to handle. Moreover, bacteria show the additional advantage of not having restrictions on the number or dimension of heterologous genes to be inserted into their genome, in contrast to viruses which encounter limits in the capacity to encapsulate foreign DNA.

Both attenuated pathogenic and commensal bacteria have been engineered to carry and deliver heterologous antigens to mucosal surfaces. Attenuated pathogenic strains are particularly effective since they are well adapted to mucosal surface environments and *Mycobacterium bovis* BCG [35,36], *Listeria monocytogenes* [37,38], *Salmonellae* spp. [39,40], and *Shigellae* spp. [41] are capable of eliciting immune responses against important viral, bacterial, protozoan, and metazoan pathogens in animal models. However, they are unsuitable for vaccinating individuals such as infants, the elderly, or immunocompromised patients because of their residual virulence [42,43].

In this regard, probiotic bacteria have been extensively studied as safe and effective vaccine candidates. Microorganisms such as lactic acid bacteria (LAB) or *Bacillus* spp. are generally recognized as safe (GRAS) bacteria and represent major microorganisms used for probiotic purposes [43]. Vaccines based on LAB, especially *Lactobacillus lactis*, *Lactobacillus plantarum* [44,45], and *Lactobacillus casei* [46], have shown promising beneficial effects in prophylactic or therapeutic probiotic-based vaccines against emerging respiratory viruses, including SARS, influenza viruses [47], and more recently, SARS-CoV-2 [48,49].

## 2. Bacterial Spores as a Mucosal Vaccine Delivery System

Bacterial spores have been proposed for the oral delivery of the COVID-19 vaccine. In particular, *Bacillus subtilis* recombinant spores expressing the SARS-CoV-2 spike protein receptor-binding domain have been developed and shown to increase specific neutralizing antibodies when orally administrated to mice or healthy human volunteers [50,51]. The utilization of *B. subtilis* spores to display heterologous antigens has been extensively reported since its first successful application in 2001 by Isticato et al., and will be the focus of this review [52].

### 2.1. Bacterial Spores, Sporulation, and Intestinal Life Cycle

Bacterial endospores are metabolically inactive cells produced by Gram-positive bacteria belonging to the genera of aerobic or facultative anaerobic Bacilli and the strictly anaerobic Clostridia [53]. These bacteria can adapt to environmental changes through different survival mechanisms, such as competence, biofilm production, cannibalism, or sporulation [54,55]. This latter process is induced when the environmental conditions do not allow cell growth or survival and the production of endospores (hereafter referred to as a “spore”) occurs. Sporulation has been widely investigated in *Bacillus subtilis*, the model system for spores [53,54,55,56].

During sporulation, the vegetative cell undergoes morphological, biochemical, and physiological changes, leading to the development of two different but genetically identical cells: the mother cell and the forespore (Figure 3). This differentiation event is possible thanks to distinct gene expression programs in the two cell types [56,57,58].

A mature spore released into the environment is one of the most resilient cell types in nature, able to survive extreme stresses such as extreme pH and temperatures, dehydration, ionizing radiation, mechanical abrasion, toxic chemicals, and hydrolytic enzymes [59,60,61]. Moreover, spores can persist in this dormant state for several decades, perhaps even longer, until the environmental conditions become favorable again [62].

Despite its dormancy, a spore can quickly germinate and resume vegetative growth in response to water, nutrients, and favorable environmental conditions (Figure 3) [62].

The exceptional longevity of the spore in the environment is the main reason for the ubiquitous distribution of these organisms in nature, being found in soil, air, fermented foods, and the human gut [63,64].

In particular, ingested spores of *B. subtilis* have been shown to pass safely through the stomach, and then germinate and proliferate in the upper intestine, triggered by the nutrient-rich environment of the small intestine [62,63,64,65] (Figure 4). Then, germination-derived cells can sporulate in the lower part of the intestine, thus performing their entire life cycle in the animal gut [65,66]. Several works have shown that many *Bacillus* species, including *Bacillus clausii*, *Bacillus pumilus*, *Bacillus polyfermenticus*, and *B. subtilis* contribute to the composition of gut microbiota and so maintain host health by performing nutritional, immunological, and physiological functions [66,67,68,69,70].

As observed in transmission electron micrographs, the spores of *B. subtilis* appear to be formed from several layers surrounding an innermost part, the core (Figure 5) [71]. The core contains a partially dehydrated cytoplasm, a copy of the chromosome, RNAs, and inactive enzymes [53,54,55,56,57,58,59,60,61,62,63,64,65,66,67,68,69,70,71]. The highly dehydrated state of the core is due to the replacement of water with Ca^2+^-dipicolinic acid and is responsible for the heat resistance of the spores. The tolerance to damage from UVA radiation is due to small acid-soluble proteins (SAPS) that maintain the DNA in a compact state. The core is protected by the cortex, a thick peptidoglycan-like structure, and the coat, a multilayered proteinaceous layer [72]. In turn, the coat is formed from four concentric sublayers: the basement layer, inner coat, outer coat, and crust, which is made of glycoproteins (Figure 5) [73].

### 2.2. Spore Surface

In *B. subtilis*, the coat represents the spore surface and comprises over 80 proteins, representing ∼25% of the total spore proteins. All the genes coding for the Cot proteins are expressed in the mother cell under the control of two sporulation-specific sigma factors of the RNA polymerase, SigE and SigK, and at least three transcription regulators, SpoIIID, GerE, and GerR [74]. This complex transcriptional regulation ensures the synthesis of each spore surface component at the correct time during sporulation [56,58]. Post-translational modifications, such as phosphorylation, cross-linking, or glycosylation, occur after their synthesis by enzymes specific to sporulation. While cross-linking, including disulfide cross-links and dityrosine cross-links, contributes to the strength of the coat, the glycosylation of coat proteins determines the hydrophobicity of the spore and, therefore, its adherence properties [75,76,77].

Finally, in late-stage spore development, the assembly of the coat components involves various coat proteins with morphogenetic activity, i.e., proteins that do not affect the synthesis of other coat proteins but are essential for the organization of the coat into the multi-layered final structure [78,79,80,81,82]. In particular, some coat proteins are glycosylated in the late-stage spore development and assembled on the surface of the forespores, forming the crust layer [77,83].

Several studies have highlighted the ability of some coat components to self-assemble and form remarkably robust structures characterized by hexagonally symmetric semipermeable arrays [84]. The final result of coat development is a complex layer arranged into two major layers: an outer, darker layer with coarse striations, and a lighter, inner layer with a lamellar appearance [71]. The two other sublayers, the basement layer and the crust, under the inner coat and external to the outer coat, respectively, complete the proteinaceous spore surface. The physical–chemical properties of the proteins and carbohydrates which compose the coat of the mature spore give it a negative charge and a relative hydrophilicity to the spore surface [85], as suggested by the altered electric charge and hydrophilicity of mutant spores lacking some coat components [77,83,86].

Spores of *Bacillus megaterium* or *Bacillus cereus* and its close relatives *Bacillus anthracis* and *Bacillus thuringensis* are surrounded by a more complex crust layer, called the exosporium [87]. The exosporium consists of a basal layer of glycoproteins surrounded by an external nap of hairlike protrusions, and, unlike the crust of *B. subtilis* spores, is loosely attached to the coat [84]. Among *Bacillus* species, the spore’s surface differs not only in the presence or absence of the exosporium but also in the diversity of the proteins that comprise the coat. Genome sequence analysis of the *Bacillus* species has highlighted that less than one half of the known *B. subtilis* coat protein genes have recognizable orthologues in its close relatives. In the genomes of *Clostridium* spp., the *B. subtilis* coat genes are even less conserved [88]. Despite the spore surface’s physical–chemical properties and structure varying from species to species, the coat of bacterial spores, by itself or in combination with the exosporium, plays a crucial role in spore protection and the initiation of germination [89].

### 2.3. Bacterial Spore as a Recombinant Vaccine Platform

The rigidity and compactness of the coat, along with spore stability and resistance, have suggested the use of spores as a display system for the formulation of mucosal vaccines. A genetic system to engineer the surface of the *B. subtilis* spore was developed in a pioneering study in 2001, and model antigens were efficiently displayed on the spore surface using the coat components as anchoring motifs [52]. Since then, various antigens have been successfully exposed on the surface of recombinant spores of *B. subtilis* and other spore formers, and shown to induce antigen-specific, protective immune responses in mucosally immunized mice [90,91].

With respect to the systems based on microbial cells or phages described above, the spore-based approach provides at least four main advantages:(1)The extreme stability due to the well-documented resistance of the bacterial spore to high temperatures, acidic pH, and the presence of chemicals and enzymes [52,53]. Guaranteeing the high stability of the vaccine carrier system from production to administration to the patient is a crucial requirement of vaccine development. *Bacillus* spores are more stable than vegetative cells during the processing and storage stages of commercial preparations, making them a suitable candidate for vaccine formulations [92]. Moreover, stability at extreme temperatures is preferred in the development of mucosal vaccines, mainly for those intended for use in developing countries, where poor distribution and storage conditions are the main limitations [6,91,93];(2)The exceptional safety record of several spore former species used worldwide in probiotic preparations for human and animal use, as dietary supplements and growth promoters [66,67,68,69,70,92,94]. As mentioned above, several spore former species are part of the animal gut microbiota, which have a role in the development of the immune system, protection against intestinal pathogens, induction of cytoprotective responses, and of anti-oxidative stress responses in epithelial cells (Figure 4) [64,65,66,94]. This safety record is an essential requirement if the display system is intended for the delivery of antigen molecules to human mucosal surfaces;(3)All known coat proteins are synthesized in the mother compartment during sporulation [54,56,72]. Consequently, coat components and the antigens fused to them do not need to undergo a cell wall translocation step to expose the recombinant proteins externally, thus overcoming the size limitation often encountered with cell-based display systems [52,91,95];(4)Several large proteins have been successfully displayed on the spore surface without affecting the spore’s structure, resistance properties, or germination [52,95]. These results are due to the compactness of the coat and the dispensable role of the coat components used as carriers [52,95].

### 2.4. Spore-Based Vaccine Design Strategy

In recombinant vaccines, the immunogenicity of an antigen depends on several factors, such as its surface exposition and the fold stability of the protein chimer.

The conformational stability of the antigen, for example, affects its intracellular fate during processing in antigen-presenting cell APC (from the uptake to the final presentation of antigenic peptides) and downstream events, such as the stimulation of B and T cells or the differentiation of T cells into effector cells [96,97].

The strategy to display antigens on the spore surface and the choice of anchor protein are crucial for the success of the recombinant vaccine. Efficient coat-carrier proteins have to meet specific criteria, such as having a solid anchoring motif to ensure the attachment of the fusion protein on the spore surface and a domain to correctly display the epitopes. Moreover, the coat component should ensure the conformational stability of the recombinant protein to natural stresses, i.e., pH, temperature, or redox environment. Maintaining antigen folding is crucial to avoid the peptide backbone being accessible to proteases, which are abundant on mucosal surfaces and in the extracellular matrix [97,98]. Finally, the fusion of the antigen to the spore component should not affect any of the functions or proprieties of the first one. Therefore, fusions at the *N*-terminus, C-terminus, or interior of the Cot protein (sandwich fusions) are often constructed with the antigen to obtain an efficient display (Figure 6). An *N*-terminus fusion is preferred when the Cot protein has the anchoring domain in its C-terminus part or when the epitope is present at the *N*-terminal of the antigen. On the other hand, a C-terminus fusion could be selected if the anchoring domain is in the *N*-terminal of the coat protein [99,100].

#### Anchor Proteins

Most antigens exposed on the spore surface have been anchored to the crust-associated component CotG [10,11,12,13,14,15,16,17,18,19,20,21,22,23,24,25,26,27,28,29,30,31,32,33,34,35,36,37,38,39,40,41,42,43,44,45,46,47,48,49,50,51,52,53,54,55,56,57,58,59,60,61,62,63,64,65,66,67,68,69,70,71,72,73,73,74,75,76,77,78,78,79,80,81,82,83,84,85,86,87,88,89,90,91,92,93,94,95,96,97,98,99,100,101,102,103] or the outer coat proteins CotB and CotC [78,79,81,102]. The surface localization of the three spore coat proteins and their abundance have ensured the exposure of many heterologous proteins (Table 1).

In particular, CotB was used in the first attempt at spore surface display technology, fusing the non-toxic C-terminal fragment of the tetanus toxin (TTFC), a highly immunogenic and well-characterized peptide encoded by the tetC gene of *Clostridium tetani* [52]. The correct timing of the expression of CotB-TTFC during sporulation was guaranteed by placing the chimera protein under the *cotB* promoter, while the chromosomal integration of the *cotB*-*tetC* fusion ensured the genetic stability of the construct (Figure 6) [52].

Since the anchor motif of CotB was not yet identified, the *tetC* gene was cloned at the 3′ and 5′ of the *cotB* gene and inserted in an internal region, obtaining a sandwich fusion. Western, dot blot and fluorescent-activated cell sorting analyses showed that similar amounts of TTFC were expressed on the surface of the three recombinant spores, suggesting that the fusion position did not affect the efficiency of the display system. Moreover, it was estimated that a single spore exposed more than 1.5 × 10^3^ TTFC molecules on its surface [52]. When orally or nasally administered to mice, the recombinant spores induced the production of TTFC-specific fecal sIgA and serum IgG and protected mice in a challenging experiment with purified tetanus toxin [52,104].

After these first studies, other antigens have been successfully displayed on *B. subtilis* spores using CotB as an anchor protein, such as domains 1b-3 and 4 of the protective antigen (PA) of *B. anthracis* [95,105], the C-terminal part of the alpha toxin of *Clostridium perfringen* [106], the UreA protein of *Helicobacter acinonychis* [107], the carboxy-terminal repeat domains of toxins A and B (TcdA-TcdB) of *Clostridium difficile* [108], and the VP28 protein of the white spot syndrome virus [109] (Table 1).

The strategy initially developed to use CotB as an antigen-displayed protein has been followed for other coat components. For example, TTFC and the subunit B of *Escherichia coli* labile toxin (LTB) [110] and TcdA-TcdB of *C. difficile* [108] have been exposed on spores using the 12 kDa CotC as an anchor motif (Table 1). In all cases, the coat-protein-coding gene and its promoter have been cloned in-frame with the antigen coding sequence to construct a translational fusion followed by the chromosomal integration of the gene fusions [111]. Contrary to what was observed for CotB, the antigen display efficiency of the CotC-based chimera depends on the position of insertion of the heterologous part, and a five-fold increase in the efficiency of the display was observed when TTFC was at the *N*-terminal end of CotC rather than at its C-terminal end [110,111]. In recent years, mucosal spore-based vaccines against *Clonorchis sinensis* were proposed using CotC as the anchoring protein, since the increase in food-borne infectious diseases caused by the ingestion of contaminated raw or undercooked fish [112]. Several liver fluke antigens, such as CsTP22.3, TP20.8, CsPmy, and CsLAP2, were successfully displayed on spore surfaces [113,114,115,116,117]. When the recombinant spores were intragastrically administrated to mice, an immunological response was observed with specific IgGs in sera and an sIgA increase [64,113,115].

CotG also was successfully used as a carrier to display proteins such as UreA of *H. acinonychis* [107] and a fragment of the *C. difficile* flagellar cap FliD protein [99]. Interestingly, when the biotin-binding streptavidin was fused to CotG, the recombinant proteins could still form the native tetrameric form on the spore surface [118,119]. Several antigens, i.e., viral antigens, are multimer and can only induce an immune response in this complex form. Therefore, the spore’s ability to display the antigen in a multimeric form that mimics epitopes’ natural presentation could ensure protective responses.

In recent work, the transmissible gastroenteritis virus spike (TGEV-S) protein was anchored on spores by CotG [87]. When vaccinated with the recombinant spores, suckling piglets showed an elevated specific S-IgA titer in feces and IgG titers and neutralizing antibodies in serum. Moreover, the animals were protected in TGEV challenge experiments. Altogether, these results highlight the considerable potentiality of CotG as carrier proteins for the formulation of mucosal vaccines [120].

Two other coat proteins have been used as anchor motifs to expose antigens, CotZ and CgeA, both crust components [77,121]. CotZ was proposed as an anchor motif for the first time by Negri et al., to expose a fragment of the *C. difficile* FliD protein [84]. Despite CotZ localization in the crust, the outermost layer of the spore, the surface display efficiency of the recombinant FliD protein was similar to that measured when the heterologous protein was fused to CotB, CotG, and CotC [99]. These results suggested that the exposition of the chimera protein on the spore surface does not necessarily depend on the localization of the coat protein used as carrier protein but also the passenger protein and the properties of the obtained fusion.

Finally, CgeA was used to express the *H. pylori* NCTC 11637 cytotoxic-associated gene A protein (CagA) on the spore surface [99].

**Table 1 biomolecules-13-00947-t001:** Coat proteins of *B. subtilis* proposed as anchor motifs for antigens displaying on the spore surface.

Anchor [Refs]	Target Pathogen	Antigen	Fusion Method	Linker	Application
**CotB**					
[52]	* **Clostridium tetani** *	C-term fragment of the tetanus toxin, **TTFC**	C-term, *N*-term, sandwich	-	Oral vaccination for tetanus
[99]	* **Clostridium difficile** *	fagellin protein, **FliD**	C-terminal	GGGEA; AAKGGG	*C. difficile* oral vaccine
[108]		C-term repeat domains of toxins A and B, **TcdA-TcdB**	C-terminal	-	
[106]	* **Clostridium perfringens** *	C-term of alpha toxin gene, **Cpa247-370** fused to the GST gene	C-terminal	-	Oral and Nasal Vaccine against necrotic enteritis
[95,105]	* **Bacillus anthracis** *	anthrax-protective antigen, **PA**	C-terminal	-	Anthrax vaccine
[107]	* **Helicobacter acinonychis** *	urease subunit alpha, **UreA**	C-terminal	GGGEAA; AKGGG	Anti-*Helicobacter* vaccine
[121]	* **Helicobacter pylori** *	vacuolating cytotoxin A, **CagA**	C-terminal *N*-terminal	GGGGS	Anti-*Helicobacter* vaccine
[122]	* **Mycobacterium tuberculosis** *	immunodominant secretory antigen, **MPT64**	C-terminal	-	Nasal Vaccine against tuberculosis
[100]	* **Streptococcus mutans** *	truncated **P1** protein	*N*-terminal	NR	*S. mutans* vaccine
[109]	**White spot syndrome virus**	major envelope proteins, **VP28**	C-terminal	-	Oral vaccine for shrimps
[123]	**Influenza virus**	ectodomain of influenza virus **M2** protein	C-terminal	-	Oral Influenza vaccine
[124]	**Adjuvant**	human **IL-2**	C-terminal		Adjuvant to *H. pylori* vaccine
**CotC**					
[110]	* **Clostridium tetani** *	C-term fragment of the tetanus toxin, **TTFC**	C-terminal *N*-terminal	-	Oral vaccination for tetanus
[108]	* **Clostrium difficile** *	C-term repeat domains of toxins A and B **TcdA-TcdB**	C-terminal	-	*C. dìfficile* oral vaccine
[110]	* **Escherichia coli** *	heat-labile enterotoxin B, **LTB**	C-terminal *N*-terminal	-	*E. coli* vaccine
[125]	* **Salmonella serovar pullorum** *	outer membrane protein (porin), **OmpC**	C-terminal	-	*Salmonella* vaccine
[95,105]	* **Bacillus anthracis** *	anthrax-protective antigen, **PA**	C-terminal	-	Anthrax vaccine
[67]	* **Helicobacter acinonychis** *	urease subunit alpha, **UreA**	C-terminal	-	Anti-*Helicobacter* vaccine
[124]	* **Helicobacter pylori** *	urease subunit beta, **UreB**	C-terminal	-	Oral vaccine for *H. pylori*
[121]		vacuolating cytotoxin A, **CagA**	C-terminal *N*-terminal	GGGGS	
[126]		cholera toxin B subunit, **CTB** and **UreB**	C-terminal	-	
[112]	* **Clonorchis sinensis** *	tegumental protein 20.8 kD, **TP20.8**	C-terminal	-	Liver flukes vaccine
[112]		tegumental protein 22.3 kDa, **CsTP22.3**	C-terminal	-	
[113]		cysteine proteases, **CsCP**	C-terminal	-	
[114]		leucine aminopeptidase 2, **CsLAP2**	C-terminal	-	
[115]		enolase	C-terminal	-	
[116]		paramyosin antigen, **CsPmy**	C-terminal		
[117]		serpin, **CsSer-3**	C-terminal		
[127]	* **Schistosoma japonicum** *	26 kDa full- length GST protein, **SjGST**	C-terminal	-	Liver flukes oral vaccine
[128]	**grass carp reovirus**	major outer capsid protein, **VP4**	C-terminal	-	Grass carp reovirus vaccine
[129]	**White spot syndrome virus**	major envelope proteins, **VP28** and **VP62**	C-terminal	-	Oral vaccine for shrimps
[130]	* **Bombyx mori** *	nucleopolyhedrovirus, **GP64**	C-terminal	-	*Bombyx mori* vaccine
**CotG**					
[107]	* **Helicobacter acinonychis** *	urease subunit alpha, **UreA**	C-terminal	-	Anti-*Helicobacter* vaccine
[99]	* **Clostridium difficile** *	fagellin protein, **FliD**	C-terminal	GGGEA AAKGGG	Oral vaccine against *C. difficile*
[121]	* **Helicobacter pylori** *	vacuolating cytotoxin A, **CagA**	C-terminal *N*-terminal	GGGGS	Anti-*Helicobacter* vaccine
[120]	**Transmissible gastroenteritis virus**	transmissible gastroenteritis virus spike, **TGEV-S**	C-terminal		Transmissible gastroenteritis vaccine
[118]	**-**	Streptavidin	C-terminal	GGGGS	-
**CotZ**					
[99]	* **Clostridium difficile** *	fagellin protein, **FliD**	C-terminal	GGGEA AAKGGG	Oral vaccine against *C. difficile*
[107]	* **Helicobacter acinonychis** *	urease subunit alpha, **UreA**	C-terminal	GGGGS	Anti-*Helicobacter* vaccine
[131]	* **Helicobacter pylori** *	vacuolating cytotoxin A, **CagA**	C-terminal *N*-terminal	GGGEAAAKGGG	Anti-*Helicobacter* vaccine
**CgeA**					
[121]	* **Helicobacter pylori** *	vacuolating cytotoxin A, **CagA**	C-terminal *N*-terminal	GGGEAAAKGGG	Anti-*Helicobacter* vaccine

### 2.5. Strategies for Optimizing Antigen Exposure on Spore Surface 

Several strategies have been developed to augment the efficiency of spore-based mucosal vaccines since spores were genetically manipulated in 2001.

#### 2.5.1. Linker Peptides to Increase Stability and Flexibility of Recombinant Protein

To ensure the folding structure of the exposed antigens, different linker peptides of 5–11 residues were introduced between the antigen and the Cot protein, working as a carrier [132,133]. The linker can affect the exposed antigen’s immunogenicity by increasing the chimeric proteins’ stability or flexibility. For example, the display efficiency of CgeA-CagA and CotB-UreA was remarkably increased when a linker was added between the antigen and the anchor proteins [107,121,131,133].

Several linker libraries with different properties have been utilized, and promising results have been created by inserting flexible linker peptides. For example, the insertion of GGGEAAAKGGG between the C-terminus of the anchor protein and the *N*-terminus of the exogenous protein UreA is more efficient compared with using GGGGS as the linker. However, when the latter linker was inserted between CotB and UreA, the recombinant proteins failed to be displayed on the spore surface, probably because the short linker did not allow the formation of any secondary structure, making the chimera unstable. Similar results were obtained with FliD when expressed using the same approach [99]. Therefore, the authors speculated that an α-helix could be formed in the GGGEAAAKGGG peptide chain but not in a shorter linker such as GGGGS, favoring the antigen display on the spore surface [107,133].

#### 2.5.2. Multi-Antigen Spore-Based Mucosal Vaccine

For vaccine development, the crucial point is to activate multiple immune cell types to be effective against aggressive pathogens. Using spores as a vaccine delivery system can expose different molecules, as well as multiple antigens or adjuvants, on the same spore. The deep knowledge of the genetics of *B. subtilis* has allowed the identification of different integration loci, such as *thrC*, *pyrD*, and *gltA* loci, to insert the chimeric proteins into its chromosome [133,134]. The integration of heterologous constructs into the target genes does not perturb the sporulation rate and the spore structure and resistance [133,134]. Using alternative integration loci gives strength and flexibility to the design of different fusions on the spore surface simultaneously. For instance, a higher number of exposed molecules can be achieved via the co-expression of the same antigen anchored to different coat proteins on the spore surface. A collection of 16 integrative plasmids has been created for ectopic integration using the five mentioned cot proteins as anchor motifs. In particular, plasmids for obtaining *N*-terminal and C-terminal fusions have been constructed for CotB, CotC, and CotG, while integrator vectors have been designed to produce only C-terminal fusions for CotZ and CgeA [121].

Another advantage of the extreme versatility of spores as a vaccine-display system is the possibility of administering a unique spore presenting different antigens on the same surface or using a mixture of spores singularly expressing antigens, for immunization. Immunization using different types of recombinant spores ensures a perfectly dosed amount of each antigen or adjuvant. Hinc and collaborators showed that recombinant spores expressing a fragment of the *H. acinonychis* UreB protein elicit a more robust cellular immune response in orally immunized mice when co-administered with spores expressing IL-2 [135].

#### 2.5.3. Non-Recombinant Display of Antigens on the Spore Surface

The release of genetically modified microorganisms into nature and their clearance can sometimes be a drawback. To overcome this issue, a non-recombinant to display antigens on the spore surface was proposed by Huang and collaborators in 2010 [136], and various model antigens have been efficiently displayed [137,138,139,140,141]. This approach uses spores’ negatively charged and hydrophobic surface layer to adsorb foreign proteins, and is more efficient at acidic pH [85,142,143,144].

Antigen-adsorbed spores have been used to immunize mice mucosally, and protective humoral and cellular immune responses have been measured [141]. For example, a report by Isticato et al. showed that LTB of *E. coli* was present on the spore surface in its pentameric native form when adsorbed on the spore surface, improving the antigen recognition by its natural receptor and the activation of a proper immune response [138]. Remarkably, killed or inactivated spores, unable to germinate, appeared as effective as live spores in stably displaying the adsorbed proteins. Although the molecular details of spore adsorption are still not totally understood, it is clear that (1) spore-adsorbed molecules are more stable than unbound, free molecules at both high temperatures and low pH values, suggesting that the interaction with the spore stabilizes and protects the heterologous protein [143]; (2) some mutant spores, either lacking or strongly altered in their outer surface structures, are more efficient than their isogenic wild-type strain in adsorbing antigens [144]; and (3) adsorbed molecules are stably attached to the spore surface and can be detached only by drastic treatments [144].

Since the remarkable progress achieved in spore physiology, recombinant and non-recombinant spore-based vaccines have been improved to ensure higher immune responses. For instance, the observation that *B. megaterium*, *B. cereus* [87], and *B. subtilis* [145] spores, produced under different laboratory conditions, have structurally different coat layers has suggested an innovative approach to modulating the number of exposed antigens. In particular, in *B. subtilis*, CotB, CotG, and CotZ are more abundantly represented when spores are produced at low temperatures (25 °C) than higher ones (37–42 °C). Others, such as CotC and CotU, show an opposite trend. These findings have been exploited to control the display of heterologous proteins on spores [146].

## 3. Conclusions

Mucosal infections are a major worldwide health concern, and it is generally accepted that mucosal vaccination strategies that can block infection at the point of entry would be preferable to other prevention approaches. However, relatively few mucosal vaccines are still available, mainly because of the need for efficient delivery systems. Recombinant bacterial spores displaying a heterologous antigen have been shown to induce protective immune responses and have been proposed as a mucosal delivery system. Several reasons support the use of spores as a vaccine delivery system: the remarkable and well-documented resistance of spores, which ensures the high stability of the display system, and the safety record of many spore-forming species, which makes spores of these species ideal candidates as vehicles to transport molecules to the surface of human mucous membranes.

Significant progress has been made regarding the understanding of spore biology in recent decades, and this new understanding of these biological mechanisms has been exploited to benefit the health of both animals and humans. Advances in the understanding of spore structure have allowed, at the same time, the improvement of spore-based display systems, with the development of different efficient approaches. Several researchers in this field have proposed combining the recombinant and non-recombinant approaches to produce single spores displaying a protein fused to a spore coat protein and a second molecule adsorbed. Potocki et al. (2017) have increased the immune response against *C. difficile* in mice via the intranasal administration of recombinant IL-2-expressing spores previously adsorbed with the antigen FliD. Such dual-display technology can be useful both to elicit a higher immune response, exposing an antigen and adjuvant on a single spore simultaneously, and for the specific delivery of heterologous molecules. Nguyen et al. (2013) expressed CotB-streptavidin on a spore surface and bound it to a biotinylated antibody recognizing specific tumor cells. The authors showed that the recombinant spores adsorbed with a mitotic inhibitor used in cancer therapy were able to bind HT 29 colon cancer cells and thereby deliver the drug to the cells.

Finally, considering the proven advantages, such as safety, low cost, flexibility and high-performance, of using spores for antigen display, it would be of high interest to test this delivery system in more advanced clinical trials for a future formulation of safe, economic, and targeted mucosal vaccines.

## Figures and Tables

**Figure 1 biomolecules-13-00947-f001:**
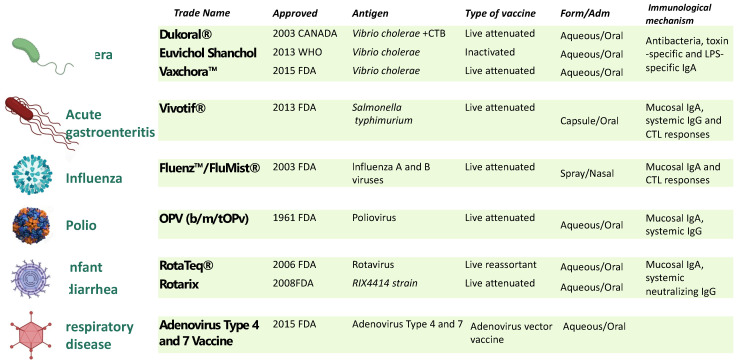
Licensed mucosal vaccine formulations for human use. Nine mucosal vaccines are currently being used against cholera caused by *V. cholerae O1* of either serotype (Inaba, Ogawa) or biotype (El Tor, Classical) [7]; typhoid fever, caused by *Salmonella enterica* serovar *Typhi* [8]; influenza caused by influenza A subtype viruses and type B viruses [9]; poliomyelitis caused by Poliovirus [10]; infantile gastroenteritis (diarrhea and vomiting) caused by *rotavirus* infections (Types G1, G2, G3, G4, and G9) [11]; and adenovirus [12]. To date, live attenuated and inactive vaccines have proven to be the most effective platforms for developing mucosal vaccines. Excluding the influenza vaccines that use the intranasal route, the remaining eight vaccines are all orally administrated. CTB: cholera toxin B subunit; LPS, lipopolysaccharide.

**Figure 2 biomolecules-13-00947-f002:**
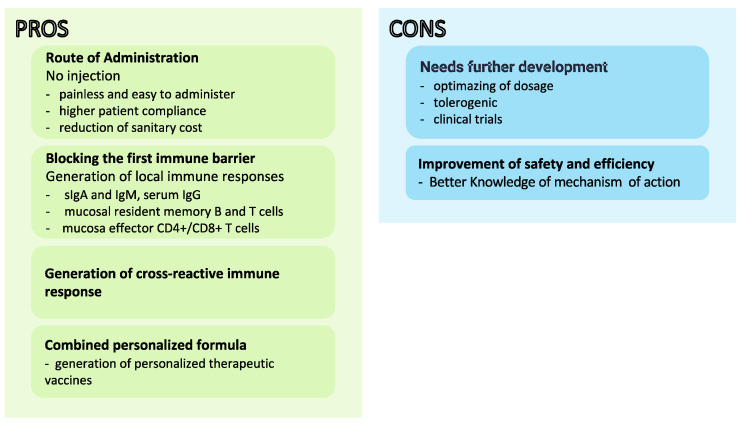
Advantages and disadvantages of mucosal vaccine development.

**Figure 3 biomolecules-13-00947-f003:**
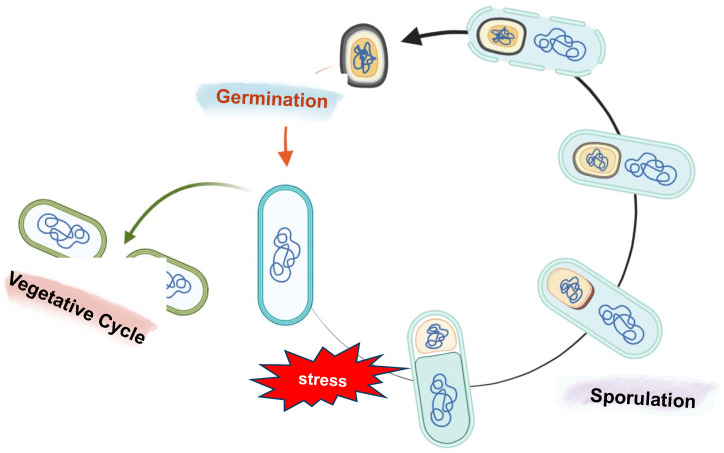
Schematic representation of sporulation and germination in *B. subtilis*. The sporulation process starts with the formation of an asymmetrical septum that produces two different compartments: a bigger one called the mother cell and the forespore, the smaller one. Next, the mother cell engulfs the forespore, and following membrane fission at the opposite pole of the sporangium, a double-membrane-bound forespore is formed. After the engulfment and throughout sporulation, several layers are placed around the forespore: the cortex, a modified peptidoglycan, is deposited between the inner and outer forespore membranes, while the proteins of the coat and then the glycoproteins of the crust are synthesized in the mother cell and assembled on the forespore surface. In the final step, the mother cell lyses to release a mature spore into the environment. If the environmental conditions ameliorate, the spore can quickly germinate and resume vegetative growth.

**Figure 4 biomolecules-13-00947-f004:**
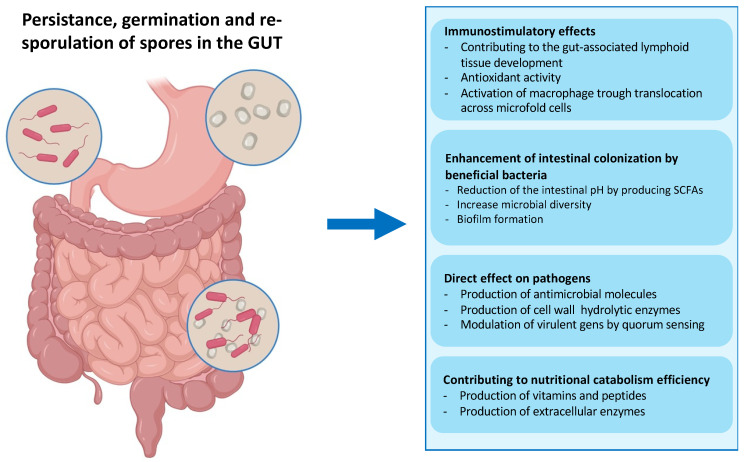
Fate of ingested bacterial spores in the gut. Spores administered via the intragastric route can transit through the stomach thanks to their extreme resistance. The low pH of the stomach causes the spores to germinate before they reach the small intestine, which is rich in nutrients. In the gut, vegetative cells help modulate the immune response and secrete hydrolytic enzymes, antioxidants, vitamins, peptides, and antimicrobial compounds, contributing to the gut microbiota balance and facilitating digestion [39]. The particularly toxic environment of the gut, including anoxia, low pH, bile salts, and an extremely high concentration of commensal bacteria, can induce the *Bacillus* cells to re-sporulate [37].

**Figure 5 biomolecules-13-00947-f005:**
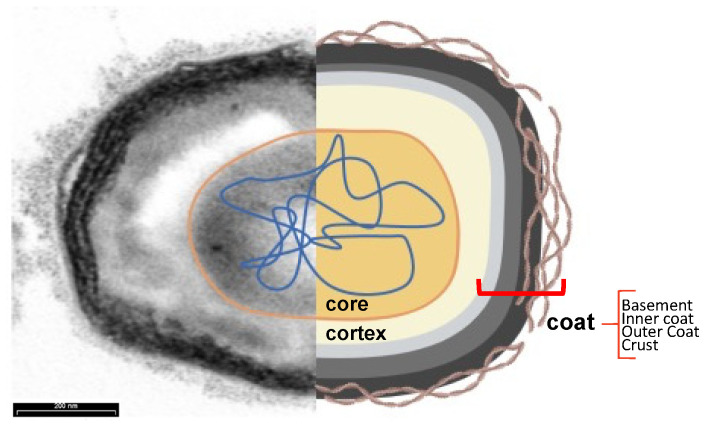
Subcellular spore structure. Transmission electron microscopy analysis (left) and cartoon (right) of a typical *B. subtilis* spore. The core (orange) contains dehydrated cytoplasm and nucleic acids (blue) and is protected by a cortex (yellow) and the four sublayers of the coat: the basement layer (light grey), the inner coat (grey), the outer coat (black), and the crust (brown filamentous), visible only after ruthenium red staining.

**Figure 6 biomolecules-13-00947-f006:**
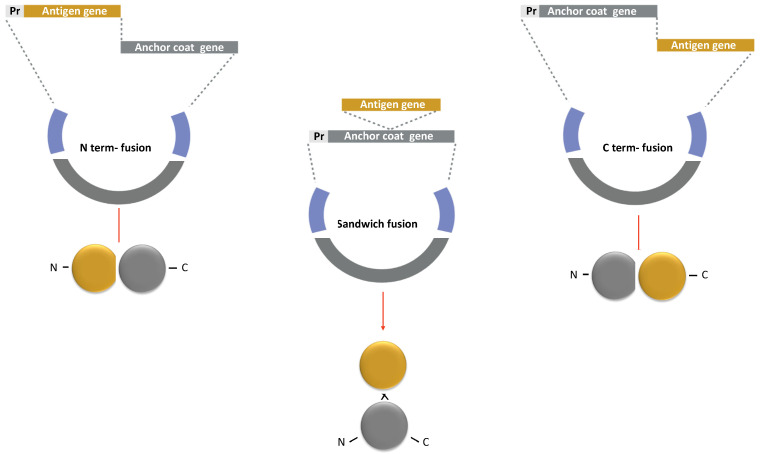
Strategy for a recombinant display of an antigen on the spore surface. The protein chimera is obtained by fusing the DNA coding for Cot protein (Anchor coat gene in grey) to the gene of the selected antigen (Antigen gene in yellow). The gene fusion is then cloned into an integrative vector between two regions of a non-essential gene of *B. subtilis* (in blue). The fusion is under the transcriptional and translational signals of the Cot protein chosen as the anchor (Pr: promoter in light grey). Similar strategies are used to obtain *N*-terminus, C-terminus, and sandwich fusions.

## Data Availability

Not applicable.

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
