# Peer review of "Bacterial Spore-Based Delivery System: 20 Years of a Versatile Approach for Innovative Vaccines"

_biomolecules, 2023, doi:10.3390/biom13060947_

Round 1

Reviewer 1 Report

The review article by R. Isticato is an interesting perspective on vaccines which target mucosal immune responses.  The focus on this review is on spore-derived vaccines and may be of interest to the journal’s readers and may expand current perceptions of vaccination strategies.  Prior to being accepted for publication several issues should be considered or revised:

1.        There are some minor language issues that should be addressed to improve the flow of the manuscript and presentation of the ideas reviewed.

a.       Line 38 change “The limited supply” to “supply-chain issues”

b.       Figure 1 legend, when listing the targets of the vaccines, I would recommend being consistent, either list all of the diseases or all of the infectious agents causing the diseases.  The list currently is a combination of the two.

c.       Line 67  It is unclear what the author intends by using the term “allied”, should this be “linked” or “conjugated”

d.       Figure 2 “mechanism” is misspelled

e.       Line 129, current accepted use is “gram-positive bacteria”, Gram is generally only used for the Gram stain…but I defer to the editors of the journal.

f.        Line 134: “bring to production” should be changed to “production”

g.       Line 179: change to “As observed in electron micrographs, the spore of B. subtilis…”

h.       Line 181:  what is “inactive” something missing?

i.         Line 209: should read “in late-stage spore development” or “in the late stage of spore development”

j.         Line 225: change “that component” to “that comprise the coat”

k.       Line 249 “strongly required” should be “necessary” or “preferred” depending upon the author’s intent.

l.         Line 355: should be “Two other coat proteins…”

m.     The large table needs a title and should be formatted differently for a better presentation if possible.

n.       Line 278: “efforted” should be “created”

o.       Line 318: “Faecal” is British English, will defer to editors of journal

p.       Line 388: should be “Multi antigen spore-based mucosal vaccine”

q.       Sections 2.5.2.-2.5.3. italicize all bacteria and genes

r.        Line 454.  This sentence is awkward and incomplete:  Perhaps could read “ Significant progress has been made regarding understanding spore biology in recent decades, and this new understanding of these biological mechanisms has been exploited to benefit the health of both animals and humans.”

2.       Line 32: subunit vaccines need to be described here as well.  Also change “treatments” to “prophylactic strategies”

3.       Line 48: Is it entirely accurate to state “mucosal immunity is not generated” or is it more accurate to state “less than optimal mucosal immunity is generated” or “low-levels of mucosal immunity is generated”…something like that? This should also have references included.

4.       In several areas, the author cites review articles for important concepts (i.e. line 80).  Citation of review articles within a review article should be limited when possible.  I ask that the author considers this concept throughout the manuscript and use primary citations whenever possible (within reason).

5.       Line 95: Are the safety issues described here legitimate (regarding reversion) or is this more of a public perception. More references supporting this concept should be applied with examples of reversions provided.

6.       Line 121.  The paragraph from 121-127 seems a bit out of place. Consider revising.

7.       Figure 3.  The font chosen in the graphic is difficult to read, consider choosing other font.

8.       Figure 4.  What is meant by “feed efficiency”? could this be “food adsorption efficiency” or “nutritional catabolism efficiency”?

some language issues can be resolved.

Author Response

We would like to thank the reviewer for their suggestions that helped to improve our manuscript.

Point-by-point reply

1a Line 38 change “The limited supply” to “supply-chain issues

Reply: DONE

1b: Figure 1 legend, when listing the targets of the vaccines, I would recommend being consistent, either list all of the diseases or all of the infectious agents causing the diseases.  The list currently is a combination of the two.

Reply: the legend of figure 1 has been modified as suggested, reporting both the disease and the infection agents.

1c: Line 67  It is unclear what the author intends by using the term “allied”, should this be “linked” or “conjugated”:

Reply: thanks for the useful correction, the “allied” term has been replaced with “conjugated”

1d: Figure 2 “mechanism” is misspelled

Reply: Figure 2 has been corrected, Thank you

1e: Line 129, current accepted use is “gram-positive bacteria”, Gram is generally only used for the Gram stain…but I defer to the editors of the journal.

Reply: the reviewer is definitely right, the "Gram" has been corrected in "gram"

1f: Line 134: “bring to production” should be changed to “production”:

Reply: the sentence has been changed

1g: Line 179: change to “As observed in electron micrographs, the spore of B. subtilis…”

Reply: thank you very much for the correction

1h: Line 181:  what is “inactive” something missing?

Reply: the reviewer is right, “enzimes” was accidentally deleted. Again thank you

1i: Line 209: should read “in late-stage spore development” or “in the late stage of spore development”

Reply: the sentence has been modified as suggested

1j: Line 225: change “that component” to “that comprise the coat”. Reply: DONE

1k: Line 249 “strongly required” should be “necessary” or “preferred” depending upon the author’s intent. Reply: “Preferred” has been used

1l: Line 355: should be “Two other coat proteins…” Reply: DONE

1m: The large table needs a title and should be formatted differently for a better presentation if possible

Reply: The title has been added and the format improved

1n.       Line 278: “efforted” should be “created”. Reply: done

1o.       Line 318: “Faecal” is British English, will defer to editors of journal.

Reply: the world has been replaced with fecal

1p.       Line 388: should be “Multi antigen spore-based mucosal vaccine”. Reply: done

1q.       Sections 2.5.2.-2.5.3. italicize all bacteria and genes.

Reply: done, We apologize for the uncorrected format.

1r.        Line 454.  This sentence is awkward and incomplete:  Perhaps could read “ Significant progress has been made regarding understanding spore biology in recent decades, and this new understanding of these biological mechanisms has been exploited to benefit the health of both animals and humans.”

Reply: thank you for the useful suggestion, the sentence has been inserted into the text

  1. Line 32: subunit vaccines need to be described here as well.  Also change “treatments” to “prophylactic strategies”.

Reply: a sentence has been added to describe the subunit vaccine, and the correction has been made

  1. Line 48: Is it entirely accurate to state “mucosal immunity is not generated” or is it more accurate to state “less than optimal mucosal immunity is generated” or “low-levels of mucosal immunity is generated”…something like that? This should also have references included.

Reply: thank you again for your suggestion. The text has been modified. The reference was inserted at the end of the paragraph (lines 48-53). I added the reference at the end of the specific sentence.

  1. In several areas, the author cites review articles for important concepts (i.e. line 80).  Citation of review articles within a review article should be limited when possible.  I ask that the author considers this concept throughout the manuscript and use primary citations whenever possible (within reason).

Reply: The bibliography has been revised following the correct suggestion.

  1. Line 95: Are the safety issues described here legitimate (regarding reversion) or is this more of a public perception. More references supporting this concept should be applied with examples of reversions provided.

Reply: I apologize for the mistake; the references were accidentally deleted and are now inserted correctly.

  1. Line 121.  The paragraph from 121-127 seems a bit out of place. Consider revising.

Reply: The paragraph was inserted to introduce the use of probiotics, such as LAB and spore-forming bacteria, as vaccine delivery systems.

  1. Figure 3.  The font chosen in the graphic is difficult to read, consider choosing other font.

Reply: the font and the size of Figure 3 has been changed.

  1. Figure 4.  What is meant by “feed efficiency”? could this be “food adsorption efficiency” or “nutritional catabolism efficiency”?

Reply: “feed efficiency” was replaced with “nutritional catabolism efficiency”, which is more correct, thank you for the advice

Reviewer 2 Report

This manuscript does a comprehensive and interesting review on the Bacterial spore-based delivery systems. However I still have few points that the author should consider:

-Line 30-32: “Since their discovery at the end of the 18th century with Jenner and then Pasteur's 30 studies, vaccinations represent the only way to control infectious diseases and induce 31 herd immunity. “ I think antibiotics also control infectious diseases, it should be rephrase taking this in consideration, or change “control” to “prevent” infectious diseases.

-Line 32-335: “A long list of treatments containing live, attenuated, or killed viruses or 32 bacteria has led to the suppression or decrease of the incidence of several fatal human 33 diseases.” Once again vaccines are not treatments, their prevent diseases. This sentence should be rephrase.

- Figure 2: “mechanism” not “meccanism

- Line 121-124: concerning the SARS-CoV-2 vaccine using spores two references are missing: Vetráková A, Chovanová RK, Rechtoríková R, Krajčíková D, Barák I. Bacillus subtilis spores displaying RBD domain of SARS-CoV-2 spike protein. Comput Struct Biotechnol J. 2023;21:1550-1556. doi: 10.1016/j.csbj.2023.02.007. Epub 2023 Feb 8. PMID: 36778063; PMCID: PMC9904849.

Katsande PM, Fernández-Bastit L, Ferreira WT, Vergara-Alert J, Hess M, Lloyd-Jones K, Hong HA, Segales J, Cutting SM. Heterologous Systemic Prime-Intranasal Boosting Using a Spore SARS-CoV-2 Vaccine Confers Mucosal Immunity and Cross-Reactive Antibodies in Mice as well as Protection in Hamsters. Vaccines (Basel). 2022 Nov 10;10(11):1900. doi: 10.3390/vaccines10111900. PMID: 36366408; PMCID: PMC9692796

Line 164: Fig.4 nor Fig. 5

Line 179: Should be Transmission Electron Microscope, since there are other types of electron microscopes, but the spore description is related to the transmission electron microscope.

Line 181: “cytoplasm, a copy of the chromosome, RNAs, and inactive [32,35].” I think something is missing at the end of the sentence.

Line 183: UVA not UA

Line 210: forespore not pre-spore

Line 221: “Spores of Bacillus megaterium or in Bacillus cereus and its close relatives Bacillus anthracis and Bacillus thuringensis are surrounded by a further layer, the exosporium [56,57]”. I don’t agree with this sentence. In fact the crust from B. subtilis correspond to the called exosporium in other Bacillus strains. Therefore, this Bacillus do not have a further layer, the exosporium. The exosporium correspond to the described above crust of Bacillus subtilis. It should be rephrase.

Line 302: At least, CotB is an outer coat protein, not crust.

Figure 6: The scheme that represent the sandwich fusion is not clear

Line 403: Other integrating vectors  are published, for example: Bartels J, López Castellanos S, Radeck J, Mascher T. Sporobeads: The Utilization of the Bacillus subtilis Endospore Crust as a Protein Display Platform. ACS Synth Biol. 2018 Feb 16;7(2):452-461. doi: 10.1021/acssynbio.7b00285. Epub 2018 Jan 19. PMID: 29284082.

Lines 435-438: “For instance, the observation that B. megaterium, B. cereus [115], and B. subtilis 435 [116] spores of several Bacillus species produced under different laboratory conditions have structurally different coat layers has suggested an innovative approach to modulating the number of exposed antigens.” This sentence does not make sense, it should be rephrase.

 General comments:

 I believe that the references throughout the manuscript should be the originals and not reviews.  For example in lines 211-213, reference 52 is given, it would be more accurate to refer the original papers, such as: Jiang S, Wan Q, Krajcikova D, Tang J, Tzokov SB, Barak I, Bullough PA. Diverse supramolecular structures formed by self-assembling proteins of the Bacillus subtilis spore coat. Mol Microbiol. 2015 Jul;97(2):347-59. doi: 10.1111/mmi.13030. Epub 2015 May 15. PMID: 25872412; PMCID: PMC4950064.

 Many times the gene name or the species name is not in italic. For example line 392, 393, 409…

 I think that it should be included some considerations about the vaccines using spores as delivery systems that have arrive to clinical trials or commercial use, if any.

Author Response

We would like to thank the reviewer for their suggestions that helped to improve our manuscript.

Point-by-point reply

-Line 30-32: “Since their discovery at the end of the 18th century with Jenner and then Pasteur's 30 studies, vaccinations represent the only way to control infectious diseases and induce 31 herd immunity. “ I think antibiotics also control infectious diseases, it should be rephrase taking this in consideration, or change “control” to “prevent” infectious diseases.

Reply: Thank you for the correct suggestion. The word “control” has been replaced with “prevent”

-Line 32-335: “A long list of treatments containing live, attenuated, or killed viruses or 32 bacteria has led to the suppression or decrease of the incidence of several fatal human 33 diseases.” Once again vaccines are not treatments, their prevent diseases. This sentence should be rephrase.

Reply: the sentence has been modified.

- Figure 2: “mechanism” not “meccanism. Reply: Done, thank you

- Line 121-124: concerning the SARS-CoV-2 vaccine using spores two references are missing: 

Replay: thank you very much, the references have been inserted

Line 164: Fig.4 nor Fig. 5. Reply: Done

Line 179: Should be Transmission Electron Microscope, since there are other types of electron microscopes, but the spore description is related to the transmission electron microscope.

Reply: Done

Line 181: “cytoplasm, a copy of the chromosome, RNAs, and inactive [32,35].” I think something is missing at the end of the sentence.

Reply: the reviewer is right, “enzimes” was accidentally deleted. Again thank you

Line 183: UVA not UA. Reply: Done

Line 210: forespore not pre-spore. Reply: Done

Line 221: “Spores of Bacillus megaterium or in Bacillus cereus and its close relatives Bacillus anthracis and Bacillus thuringensis are surrounded by a further layer, the exosporium [56,57]”. I don’t agree with this sentence. In fact the crust from B. subtilis correspond to the called exosporium in other Bacillus strains. Therefore, this Bacillus do not have a further layer, the exosporium. The exosporium correspond to the described above crust of Bacillus subtilis. It should be rephrase.

Reply: I agree with the reviewer and the sentence has been rephrased. Nevertheless, most research papers on the spore surface layer refer to crust and exosporium as two distinct structures

Line 302: At least, CotB is an outer coat protein, not crust.

Reply: It was a mistake,  the sentence has been rephrased.

Figure 6: The scheme that represent the sandwich fusion is not clear

Reply: The figure has been modified, I hope the sandwich fusion cartoon is clear

Line 403: Other integrating vectors  are published, for example: Bartels J, López Castellanos S, Radeck J, Mascher T. Sporobeads: The Utilization of the Bacillus subtilis Endospore Crust as a Protein Display Platform. ACS Synth Biol. 2018 Feb 16;7(2):452-461. doi: 10.1021/acssynbio.7b00285. Epub 2018 Jan 19. PMID: 29284082.

Reply: The paper of Bartels and collaborators is very interesting. Nevertheless, for this review, I have focused only on works describing spore display systems constructed for antigens delivery/vaccines development.

Lines 435-438: “For instance, the observation that B. megateriumB. cereus [115], and B. subtilis 435 [116] spores of several Bacillus species produced under different laboratory conditions have structurally different coat layers has suggested an innovative approach to modulating the number of exposed antigens.” This sentence does not make sense, it should be rephrase.

 Reply: thank you for the advice, it was a typo of an old version.

 General comments:

 I believe that the references throughout the manuscript should be the originals and not reviews.  For example in lines 211-213, reference 52 is given, it would be more accurate to refer the original papers, such as: Jiang S, Wan Q, Krajcikova D, Tang J, Tzokov SB, Barak I, Bullough PA. Diverse supramolecular structures formed by self-assembling proteins of the Bacillus subtilis spore coat. Mol Microbiol. 2015 Jul;97(2):347-59. doi: 10.1111/mmi.13030. Epub 2015 May 15. PMID: 25872412; PMCID: PMC4950064.

Reply: The bibliography has been revised following the correct suggestion.

 Many times the gene name or the species name is not in italic. For example line 392, 393, 409…

Reply: the corrections have been done

 I think that it should be included some considerations about the vaccines using spores as delivery systems that have arrive to clinical trials or commercial use, if any.

REPLY: As far asI know, at the moment, clinical trials of vaccines based on the Bacillus spores display system have not been performed or at least are not present in the literature.